# pyAKI—An open source solution to automated acute kidney injury classification

**Christian Porschen**[1]*, **Jan Ernsting**[2,3,4], **Paul Brauckmann**[5], **Raphael Weiss**[1], **Till Würdemann**[1], **Hendrik Booke**[1], **Wida Amini**[1], **Ludwig Maidowski**[1], **Benjamin Risse**[2,4], **Tim Hahn**[3], **Thilo von Groote**[1]

**1** Department of Anaesthesiology, Intensive Care and Pain Medicine, University Hospital Müunster, Müunster, Germany, **2** Institute for Geoinformatics, University of Münster, Münster, Germany, **3** Institute for, Institute for Translational Psychiatry, University of Münster, Münster, Germany, **4** Faculty of Mathematics and Computer Science, University of Münster, Münster, Germany, **5** Münster School of Business, FH Münster University of Applied Sciences, Münster, Germany

☯ These authors contributed equally to this work.
* c_pors01@uni-muenster.de

## Abstract

### Objective

Acute kidney injury (AKI) is a frequent complication in critically ill patients, affecting up to 50% of patients in the intensive care units. The lack of standardized and open-source tools for applying the Kidney Disease Improving Global Outcomes (KDIGO) criteria to time series, requires researchers to implement classification algorithms of their own which is resource intensive and might impact study quality by introducing different interpretations of edge cases. This project introduces pyAKI, an open-source pipeline addressing this gap by providing a comprehensive solution for consistent KDIGO criteria implementation.

### Materials and methods

The pyAKI pipeline was developed and validated using a subset of the Medical Information Mart for Intensive Care (MIMIC)-IV database, a commonly used database in critical care research. We constructed a standardized data model in order to ensure reproducibility. PyAKI implements the Kidney Disease: Improving Global Outcomes (KDIGO) guideline on AKI diagnosis. After implementation of the diagnostic algorithm, using both serum creatinine and urinary output data, pyAKI was tested on a subset of patients and diagnostic accuracy was compared in a comparative analysis against annotations by physicians.

### Results

Validation against expert annotations demonstrated pyAKI's robust performance in implementing KDIGO criteria. Comparative analysis revealed its ability to surpass the quality of human labels with an accuracy of 1.0 in all categories.

**Funding:** C.P. - is supported by the Deutsche Forschungsgemeinschaft (DFG, German Research Foundation) – 493624047 (Clinician Scientist CareerS M¨unster). T.vG. is supported by the Deutsche Forschungsgemeinschaft (DFG, German Research Foundation) – 493624047.

**Competing interests:** The authors have declared that no competing interests exist.

## Discussion

The pyAKI pipeline is the first open-source solution for implementing KDIGO criteria in time series data. It provides a standardized data model and a comprehensive solution for consistent AKI classification in research applications for clinicians and data scientists working with AKI data. The pipeline's high accuracy make it a valuable tool for clinical research and decision support systems.

## Conclusion

This work introduces pyAKI as an open-source solution for implementing the KDIGO criteria for AKI diagnosis using time series data with high accuracy and performance.

## Introduction

AKI is a common organ dysfunction in critically ill patients, affecting up to 50% of all intensive care unit (ICU) patients [1, 2]. Recently, the definition of AKI has been given clear outlines based on a set of clinical routine variables, by implementing the KDIGO-definition for AKI [3]. While this definition made it possible to investigate the epidemiology of this syndrome and the sequalae, there is currently no open software tool available to identify AKI events based on the KDIGO criteria in time series data. In recent years, several clinical databases were developed containing valuable patient data for large scale retrospective analysis and machine learning applications. [4–8]. However, using those databases for training of machine or deep learning based systems remains difficult due to missing annotation. Generating these annotations is highly time consuming and costly, as medical diagnoses have to be assigned by medical experts. Moreover, implementing algorithmic classification based on the KDIGO definition can be prone to errors, which might impact study quality. A tested, publicised tool is lacking. First efforts to develop automated annotation systems started recently with OpenSep. OpenSep, which has been developed in 2022, is one of the pipelines implementing Sepsis-3 guidelines into an open-source software, leveraging reusable and standardized concepts and implementing them in Python, a commonly used programming language [9].

So far, there has not been a single open source, ready to use pipeline software implementation, processing data into a standardized definition of AKI. While projects like the MIMIC-database [4] or the Amsterdam University Medical Center Database (AmsterdamUMCdb) [5] have recently undergone efforts to implement KDIGO criteria within their own data sets, a pipeline implementation for cross-project application defining a common standard for AKI is lacking. Furthermore, the validity of these annotations in comparison to annotations labelled by medical experts was not assessed.

Therefore, we propose pyAKI—an open-source, standardized, generalizable and clinically tested pipeline for implementing KDIGO criteria in time series data. Furthermore, we defined a data model to standardize the input required for the implementation of these guidelines (Fig 1). Additionally, we provide methods to support users in transforming their own raw data into a format usable by the pipeline. To evaluate pyAKI's performance, we created an expert annotation for a dataset extracted from the MIMIC-IV database, reviewed by domain experts, that evaluated all cases at each subsequent hourly point in time, during the ICU stay. All labels are based on the KDIGO 2012 definition and were verified by an expert panel of physicians experienced in the care of patients with AKI.

## pyAKI Data Model

**Fig 1. pyAKI data model.** PK = Primary Key, dt = Datetime, bool = Boolean Value, int = Integer Value, float = Floating Point Value.

The entire software we developed is open source and available for download (github.com/aidh-ms/pyAKI). The dataset generated for validation and the software implementation was extracted from the MIMIC-IV demo dataset [10], which is licensed under an open source license and enabled us to make our validation dataset publicly available (DOI: 10.17879/17988545762).

## Materials and methods

### AKI definition

AKI was defined applying the KDIGO criteria using urine output (UO) and serum creatinine (SCr) [11]. According to KDIGO, requirement of dialysis is defined as AKI stage 3. From a computational perspective, the KDIGO criteria classify AKI via three different pathways: First, a reduction in UO; second, an elevation of the kidney function marker SCr against a predefined baseline, which in turn can be a relative or an absolute increase; third the requirement of dialysis. According to the KDIGO criteria, a reduction in UO below 0.5 milliliters per kilogram per hour (ml/kg/h) for 6 to 12 hours was defined as AKI stage 1. Reduction in UO below 0.5ml/kg/h for more than 12 hours was defined as AKI stage 2. Reduction of UO below 0.3ml/kg/h for at least 24 hours or anuria for at least 12 hours was defined as AKI stage 3. SCr elevations have to be compared to a baseline in most cases. Our methods of defining and implementing baselines for SCr are explained below. A SCr rise relative to the baseline creatinine was classified as relative creatinine stage. An 1.5–1.9 fold increase relative to baseline SCr was defined as AKI stage 1, a 2–2.9 fold increase was defined as AKI stage 2, and a 3 fold increase

**Table 1. Staging of AKI according to KDIGO criteria.**

| AKI Stage | Serum Creatinine | Urine Output | Dialysis |
|---|---|---|---|
| 1 | 1.5—1.9 fold increase or absolute elevation of $\geq$ 0.3 mg/dl relative to baseline | < 0.5 ml/kg/h for 6–12 hours | |
| 2 | 2—2.9 fold increase relative to baseline creatinine | < 0.5 ml/kg/h $\geq$ 12 hours | |
| 3 | 3 fold increase relative to baseline creatinine or absolute elevation to $\geq$ 4 mg/dl irrespective of baseline | 0.3 ml/kg/h $\geq$ 24 hours or anuria $\geq$ 12 hours | dialysis present |

was defined as AKI stage 3. An absolute SCr increase of 0.3milligrams per deciliter (mg/dl) over baseline was classified as AKI stage 1. An absolute SCr increase over 4mg/dl irrespective of the baseline was classified as AKI stage 3. Dialysis was also considered for classifying AKI and any use of dialysis was classified as KDIGO stage 3. Since, definition of anuria varies in the literature we provide users the possibility to provide their own threshold for anuria based on body weight. If a patient does not fulfill any of the KDIGO criteria categories, he or she is classified as AKI-stage 0, meaning that no AKI is present. An overview of the AKI definition according to the KDIGO criteria is depicted in Table 1. In order to use the pipeline, users have to provide values for the weight of each patient. Users can decide for themselves whether an ideal body weight or adjusted body weight should be used.

**Creatinine baseline definition.** There are several methods to define a baseline for SCr. A baseline is needed to compare SCr values against it at each point in time. Defining a baseline for SCr might vary due to data availability, data structure, local practices and standards. To take this into consideration, pyAKI contains a variety of different methods for defining a baseline of SCr: Using the minimum, mean and first value of either a fixed time frame at the start of the time series or a rolling time frame following the classification. Both time frames can have a self defined length. Also, a fixed value, e.g. a known preoperative SCr for surgical patients can be used. Lastly we also included a method to calculate a baseline creatinine value using a modification of the Cockcroft-Gault formula, Cockcroft-Gault Creatinine Clearance:

$$\text{milliliters per minute (ml/min)} = \frac{(140 - \text{age}) \times \text{weight, kg}(\times 0.85 \text{ if female})}{72 \times \text{SCr, mg/dl}} \tag{1}$$

The Cockcroft-Gault formula is used to calculate the glomerular filtration rate (GFR) using the patients weight, gender and SCr [12].

The Cockcroft-Gault formula may be inaccurate depending on the patients weight. Therefore we further modified the formula to use the adjusted bodyweight of a patient if a height is provided for the patient [13].

$$\text{SCr, mg/dl} = \frac{(140 - \text{age}) \times \text{weight, kg}(\times 0.85 \text{ if female})}{72 \times \text{Cockcroft-Gault Creatinine Clearance, ml/min}} \tag{2}$$

Given this revised definition, it can be used to calculate an expected SCr under the assumption of a specified GFR, which is 75ml/min by default, but can be modified by the user.

## AKI data model

For automated annotation of data, a standardized input format is required. We constructed a minimal required set of variables and propose a data format consisting of three different data frames, each containing a subject identifier and timestamp for mapping to individual subjects. The required data frames are depicted in Fig 1. We expect the data frames to contain hourly

measured data of each individual subject, with no missing values. At each subsequent point in time, KDIGO criteria were applied. AKI stages were evaluated separately, according to the pathways explained in Section AKI Definition and included in the output (see Fig 1). The overall AKI stage is determined by the maximum stage in any of the three categories of the AKI definition. The output of the software contains each AKI defining criterion separately, together with the overall AKI stage at each hour of the patients ICU stay, formatted as time series. In our validation experiment, AKI stages are assessed at each point in time, specifically at every hour during the ICU stay. Keeping these temporal relations within the data enables users to use it for further analysis in terms of AKI duration, recovery or temporal relation to other events.

## Software development

The pipeline was build using Python 3.12 (Python Software Foundation, Beaverton, Oregon). Starting from the data model for AKI mentioned above, we first developed interpolation methods for the data. Users should be able to parse real world data which is often sparse. By providing standardized imputation methods with high flexibility to fit the users requirements, we ensure an end-to-end pipeline from real life data to the desired output. These imputation techniques are used to up-sample and interpolate data to hourly intervals on which the AKI stages can be determined on. They are developed with a focus on customizability, where users can provide their own desired cutoffs for interpolation when data is missing over several hours. UO is interpolated using the next observable measurement and spread it out over the missing time period. E.g. a urine bag might be read out at one point in time and again 3 hours later with 300ml urine in it. It is unlikely that the patient had no urine output and then 300ml (though it is possible, again: we are leaving this decision to the user) but instead that he produced 100ml urine per hour. The preprocessor distributes this next measurement evenly over the observed time frame. For SCr, a simple carry-forward method is employed where the last known observation is kept until a new observation occurrs. User can provide a threshold in ours that they see fit, for when the preprocessor should stop interpolating and instead accept missing values as missing data. In that case, the evaluation probes will start from scratch. Users have the flexibility to utilize pyAKI without employing imputation methods, allowing them to preprocess their data into the required hourly format according to their preferences. This ensures users have full control over their input data and acknowledges the heterogeneity of data sources.

## Validation data set

To extract clinical data for our testing setup, we used the MIMIC-IV demo version 2.2, which has been published in January 2023 [10]. This version of MIMIC is an openly accessible subset of the full MIMIC database which enabled us to share data publicly and provide a higher level of transparency in our process. The full MIMIC dataset is not open for publication and requires special training and identification to be worked with. MIMIC consists of more than 60,000 patients from the ICU of the Beth Israel Deaconess Hospital in Boston, United States of America. MIMIC-demo provides a subset of 100 patients of MIMIC-IV which are shared publicly over PhysioNet [4]. 14 patients that had an AKI and one patient not fulfilling KDIGO criteria to serve as control, according to the definition implemented in the MIMIC codebase, were sampled randomly from this data set and their SCr, UO and dialysis-status was extracted from the appropriate tables. The data was fitted according to our AKI data model as depicted in Fig 1. UO was extracted in milliliter (ml), SCr in mg/dl or converted from millimol (mmol)/deciliter (dl). The need for dialysis was derived from the established MIMIC concept [14] and

converted to a Boolean variable indicating the dialysis status of the patient. The extracted data was up-sampled to hourly intervals. Since UO data was not always recorded hourly, gaps of less than 6 hours were filled using interpolation according to our preprocessing pipeline mentioned under section 2.3. No further method of imputation was used and no missing values were left. For SCr baseline, we defined the baseline according to the official KDIGO criteria [11] as a rolling window of the last seven days before each subsequent measurement for the relative SCr criterion, and a rolling window of the last 48 hours before each measurement for the absolute SCr criterion.

Physicians labels were provided by three physicians (A., W.; B., H.; M., L.) working in the department for anaesthesiology and intensive care of the University Hospital Münster in Germany. They received a standardized initiation and instruction, which is appended in the supplements S1 File, consisting of an instruction on the KDIGO criteria and a presentation of unlabelled example data in the standardized format which is also used for pyAKI. The generated labels were then reviewed by an AKI expert panel consisting of two experienced physicians (W., T.; W., R.) from the same department, who also received the standardized initiation and instruction for KDIGO criteria and are experienced in the care of patients with AKI and in AKI research. They reviewed the human annotations by their junior colleagues and annotations generated by pyAKI while the sources of the annotations (pyAKI or human) were anonymised and resolved conflicts between the two, resulting in a finally annotated data set. The full resulting data set, including labels by physicians and the result of the conflict resolution by the senior physicians are published alongside this paper (DOI:https://doi.org/10.17879/17988545762). Not only did this method evaluate pyAKI against human performance, but it provided a standardized and labelled dataframe, which enables high quality testing in potential future development cycles. As mentioned above, AKI stages were evaluated at each hour of the ICU stay independently, keeping the temporal information within the data in order to enable users to leverage this for subsequent analysis. Therefore, each patient can be assigned with dozens and hundreds of AKI labels, depending on their length of observation. Physicians were asked to take note of the time they needed for each patient to label the data. The time needed for pyAKI to label the data was measured using the Python timeit module.

## Pipeline validation

The entire workflow of the pipeline validation is depicted in Fig 2. In order to validate the output of pyAKI, we employed a two step approach: First, every implemented probe, interpolation, class, method and function was tested thoroughly on artificial data. Tests are implemented in Python using the commonly applied unittest package, which is part of the standard Python library. All tests can be viewed and rerun locally by using our remote repository hosted on GitHub. Additionally, all tests are automatically executed when code changes happen to ensure correctness of the implemented algorithms. In the second step, the pipeline was validated using the novel validation data set. To also confront pyAKI with a larger set of patients, we then compared the output of pyAKI's KDIGO classification probes to the classifications made by the MIMIC-code implementation of the KDIGO criteria on the MIMIC-IV demo dataset. The maximum AKI stages of each patient were compared to each other. Statistical analysis was done in Python as well. We used the accuracy score to compare pyAKI's performance with the physicians labels and calculated the area under the curve of the receiver operator curve for the overall AKI stage. As we aim for high quality labels using our toolbox, accuracy scores over 95% were considered acceptable.

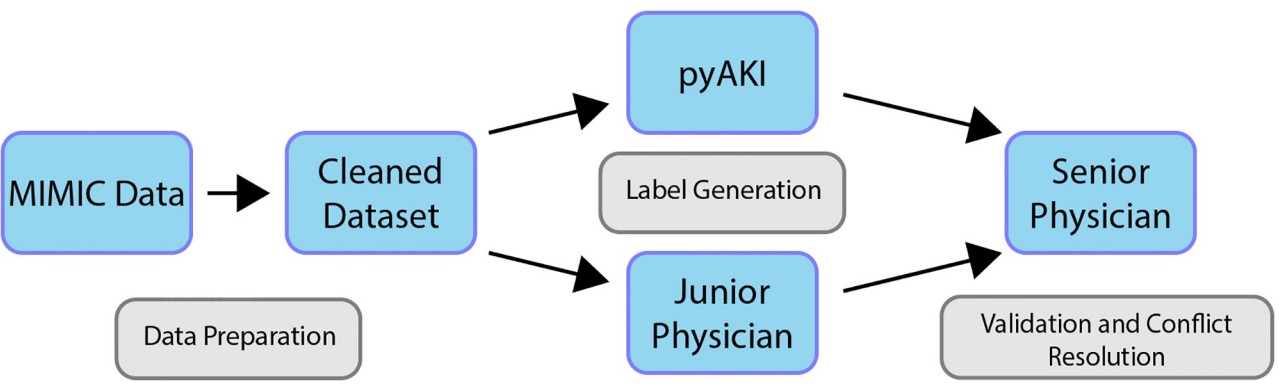

**Fig 2. Workflow of validating the pyAKI pipeline.**

## Results

PyAKI is a software package for the evaluation of time series data. Therefore, in the results, each point in time will be considered independently. In contrast to the standard expression in medical literature, we use n as measure for number of points in time, not as measure for number of patients.

### Patient cohort

15 patients were randomly drawn from the MIMIC-IV database. Overall, they included 2651 hours of patient data in the ICU. Mean length of ICU stay per patient was 176.73 hours. An overview of the descriptive parameters of the patients cohort is provided in Table 2. All patients except for one fulfilled one or more criterion for AKI classification at least at one point in time of their ICU stay. The most common AKI stage per point in time was stage 0 (n = 1623), followed by stage 3 (n = 491). A complete overview of all AKI-stage counts per point in time can be found in Table 2. The most common reason for the classification of a point in time as AKI was a reduction of urine output alone (n = 508) and in combination with dialysis (n = 179). Absolute elevations in SCr were also rather common alone and in combination with other factors (n = 269), while relative elevations in SCr alone were less likely (n = 28), yet more common in combination with other factors (n = 66). The overall sensitivity for detecting an AKI stage of any kind was 1.0 for pyAKI and 0.96 for the physicians labels. Specificity was 1.0 and 0.99, while the area under the curve was 1.0 and 0.97 respectively. The self reported cumulative time the physicians needed for the data labeling process was 972.38 minutes. It took pyAKI 0.27 seconds in order to label the data.

### Accuracy

An overview of all measured accuracies, categorized by each stage and each criterion of KDIGO classification, after comparing to the labels generated by the senior physicians is depicted in Table 2. Overall accuracy was high, both in the labels generated by physicians, as well as the labels generated by the pipeline. Accuracy across KDIGO-stages were also high (human vs. pyAKI): UO was 0.9664 accuracy score vs. 1.0 accuracy score and relative SCr elevation was 0.9996 accuracy score vs. 1.0 accuracy score. Accuracy for both dialysis-stage and absolute SCr elevation was 1.0 accuracy score both in human labelling, as well as in labels generated by pyAKI. For overall AKI-stage, accuracy score was 0.9771 for human labels and 1.0

**Table 2. Accuracy of human-assigned and pyAKI-generated labels across different acute kidney injury stages at each point in time.** AKI = Acute kidney injury, UO = urine output, SCr = serum creatinine.

| Category | AKI Label | No. of AKI per point in time | Human Accuracy | pyAKI Accuracy |
|---|---|---|---|---|
| UO | Overall | 1726 | 0.9664 | 1.0 |
| | Stage 0 | 975 | 0.9991 | 1.0 |
| | Stage 1 | 108 | 0.9630 | 1.0 |
| | Stage 2 | 299 | 0.9030 | 1.0 |
| | Stage 3 | 344 | 0.9302 | 1.0 |
| Absolute SCr Elevation | Overall | 2125 | 1.0 | 1.0 |
| | Stage 0 | 2106 | 1.0 | 1.0 |
| | Stage 1 | 223 | 1.0 | 1.0 |
| | Stage 3 | 109 | 1.0 | 1.0 |
| Relative SCr Elevation | Overall | 2438 | 0.9996 | 1.0 |
| | Stage 0 | 2311 | 0.9996 | 1.0 |
| | Stage 1 | 65 | 1.0 | 1.0 |
| | Stage 2 | 62 | 1.0 | 1.0 |
| | Stage 3 | 0 | 1.0 | 1.0 |
| Dialysis | Overall | 1914 | 1.0 | 1.0 |
| | Stage 0 | 1610 | 1.0 | 1.0 |
| | Stage 3 | 304 | 1.0 | 1.0 |
| Overall AKI Stage | Overall | 2665 | 0.9771 | 1.0 |
| | Stage 0 | 1665 | 0.9994 | 1.0 |
| | Stage 1 | 203 | 0.9704 | 1.0 |
| | Stage 2 | 306 | 0.9052 | 1.0 |
| | Stage 3 | 491 | 0.9491 | 1.0 |

for pyAKI generated labels. An overview of the accuracies across the different pathways of AKI staging is also provided in Fig 3.

## AKI characteristics

Since time series data was used continuously for determination of AKI stages both by the experts, as well as the software pipeline, an evaluation of the accuracy in time of AKI diagnosis

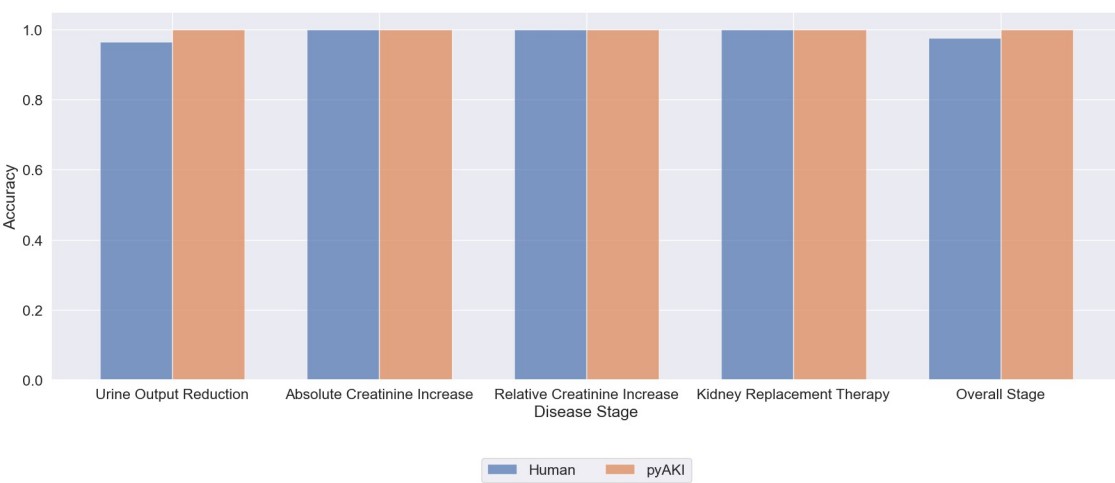

**Fig 3. Overall accuracy of human vs. pyAKI generated labels by classification method.**

is possible. Timing of first AKI diagnoses were almost identical. Only in one case discrepancies occurred between the timing of AKI diagnoses, where the UO stage was off by one hour by the physicians classification. Maximum AKI stages were evaluated with a high accuracy in all categories and the physicians labels and the labels determined by the algorithm matched in nearly all categories. Only a single case was determined incorrectly by the physicians. While the UO stage was classified correctly as stage 1, the overall stage was determined as stage 0, due to a transmission error.

## Misclassification

Errors in determining AKI stages can be grouped into three approximate categories: First, rounding errors, where errors due to rounding on digits after the decimal point occurred. Second, calculation errors, where simple miscalculation e.g. of the mean UO over a given time window lead to a misclassification in the UO stage. Third, misinterpretation or misreading of baseline values in the SCr category. Overall, most misclassifications by physicians were due to errors in calculation (n = 57). Misinterpretation of baseline values caused one error and one error was due to an error in rounding values correctly.

## MIMIC comparison

Table 3 shows the comparison of the AKI stages determined by the MIMIC-code implementation of the KDIGO criteria and the labels generated by pyAKI.

## Discussion

The proposed pyAKI software is the first standardized, validated and openly available tool for diagnosis of AKI in intensive care databases. Accordingly, the proposed data model is a first effort in harmonizing databases for subsequent analysis. Our pyAKI pipeline identified AKI patients with a high degree of agreement with ICU physicians.

PyAKI outperformed the accuracy of a group of physicians in most and equaled their performance in all metrics, including overall accuracy at each point in time, accuracy of determining the first AKI diagnosis, as well as accuracy in determining the highest AKI stage of a patient. However, these differences in accuracy were small and rarely had an effect on the overall classification of AKI stages.

Part of the superiority in performance of pyAKI opposed to the physicians might be due to the formation of the data. Tabular data is difficult to read, especially in long time series going over multiple hours or even days. This is especially true for urinary output data. Most currently used digital patient data management systems offer visualization frontends for users to facilitate easier managment for clinicians. However, our pipeline did not intend to outperform human level performance. As we have shown, human labels are of high quality and accurately represent AKI stages within time series data. By matching this quality, pyAKI is able to

**Table 3. Comparison between pyAKI's generated labels and the labels generated by the MIMIC-code implementation of the KDIGO criteria.** The maximum AKI stages of each patient were compared to each other and the quantity of labels per AKI-stage is displayed.

| AKI Stage | MIMIC labels | pyAKI labels |
| --- | --- | --- |
| 0 | 22 | 23 |
| 1 | 10 | 14 |
| 2 | 44 | 42 |
| 3 | 21 | 18 |

translate this quality to data sets on a large scale that would be resource intensive for human labelling.

Diagnosing AKI in large scale data sets is a time consuming tasks if it is done by human-generated labels and not performed algorithmically. Especially when doing so on both sparse and high frequency time series data. However, large sets of data are required where performance is dependent on the amount of available data, especially in machine learning. Past studies commonly relied on clinical labels or coded diagnoses for a patient within their hospital stay when investigating AKI. Prior studies demonstrated that such labelling is likely to create an under-representation of AKI in populations, especially for AKI stage 1 [15]. Khadzhynov et al. reported, that less than 30% of AKI episodes were transferred to the medical documentation, depending on their stage [16]. Wilson et al. also confirmed this finding in a large US population, where they even found a decreased 30 day mortality when AKI was not documented. This most likely occurred in less severe stages of illness. This relationship was reversed, when corrected for different scores of illness [17]. Even when working on the same data set, incidences of reported AKI can vary. In the EPI-AKI study, the authors were able to demonstrate that AKI across all stages occurs in more than 50%, and moderate or severe AKI occurs in more than 30% of all patients in the ICU [1]. However, some studies, especially in the field of machine learning when working with large data sets and algorithmic implementation of KDIGO stages, reported incidences that differ substantially from this benchmark [18–21]. This begs the question of why this deviation occurs and if it can all be attributed to local differences or if the algorithmic approach to classifying AKI might be flawed. Since most of the time implementations of such detection algorithms are not released with the publication, there is no way to reproduce the analysis and provide evidence for correctness of the output. There might be a lack of a standardized, well tested benchmark algorithm that can be referenced. By creating an expert-labelled validation data set and developing a toolbox upon it by testing it against that data, we aim to introduce this urgently needed standard.

In order to provide additional validation to our pipeline, we also compared the output of pyAKI's classficiation on the MIMIC-IV demo datatset to the labels generated by the MIMIC-code implementation of the KDIGO criteria. The comparison showed that pyAKI's classification was in line with the MIMIC-code implementation of the KDIGO criteria. The MIMIC-code implementation is not validated against human performance, however it is well tested and reviewed by the open source community around MIMIC and has been used in several studies [14]. This comparison provides additional evidence for the correctness of the pyAKI pipeline. Slight differences in classifications might be attributable to differences in the implementation of the KDIGO criteria, as well as differences in the data preprocessing. The MIMIC-code implementation e.g. excludes data windows where there is more missing data than the length of the window, while pyAKI interpolates the data in these cases. This might lead to differences in the classification of AKI stages, especially in stage 1, where the UO criterion is most likely to be met. As mentioned before, pyAKI is designed to be highly customizable and users can choose to use the imputation methods provided by the pipeline or preprocess their data in a different way. This allows users to adapt the pipeline to their data and preferences and ensures that the pipeline can be used in a wide range of applications.

To our knowledge, pyAKI is the first publicly available software pipeline for algorithmic determination of AKI stages on time series data. A recent study investigated the approach of algorithmic detection of AKI by applying KDIGO criteria [21]. In this study, the authors included 21045 cases of post-cardiac surgery patients from 2012 to 2022 and applied KDIGO criteria in an automated, algorithmic fashion using the programming language R (R Foundation for Statistical Computing, Vienna, Austria) [22]. However, the authors did not evaluate their calculations against human level performance or another standardized method of

classification. While detecting AKI rates of over 60% in postoperative AKI, which is close to the expected AKI rates described by Hoste et al. and Zarbock et al. [1, 2], the authors to our knowledge and to this date, did not publicise their algorithm for AKI detection which prohibits validation of their findings.

The KDIGO guidelines do not provide a clear recommendation on what qualifies as baseline SCr for defining AKI stages based on SCr and several methods are used by clinicians. To account for this, we have incorporated the most commonly used methods as mentioned under Section Creatinine Baseline Definition: Users can choose the minimum, mean or first value of a time window at the start of the time series, as well as a rolling window following the classification. The length of the window can be self defined in both cases. Users can also choose to provide a fixed value if a known baseline is used (e.g. preoperative SCr), as well as choosing a method of calculating the baseline under the assumption of a user defined GFR via the Cockcroft-Gault formula for use in pyAKI and allow the user to choose the most appropriate method based on individual preference. By providing a set of different formulas and methods for baseline calculation we intend to increase transparency across study applications. The definition of a baseline might vary due to a number of reasons. Most commonly, especially in surgical patients, a preoperative or pre-hospitalisation SCr might be the most appropriate approximation of a baseline creatinine value. This will often not be accessible though, especially when working only with ICU data. Our pipeline offers the user the possibility to be fully transparent about the exact implementation of a baseline SCr within their data. Using the Cockcroft-Gault formula as a possibility of baseline definition might be considered the most unsuitable method, as the GFR can be inconsistent and biased, especially in ICU patients, patients affected by chronic kidney disease or patients using diuretics. Therefore this possibility should be considered a last option if no other way of baseline definition is available. It might be a valid option if a study population is sure to be without patients affected by chronic kidney disease and no other method of baseline definition seems applicable.

Using Python as a programming language was an intentional decision. Other languages like R, Matlab (The MathWorks Inc., Natick, Massachusetts) or SPSS (IBM, Armonk, New York) are also commonly used in the field of medical data analysis. However, Python is the most commonly used programming language in the field of data science and machine learning. It is also the most commonly used language in the field of medical data analysis [23, 24]. By using Python, we ensure that the pipeline can be easily integrated into existing data analysis workflows. Other applications and tools that find common usage in the data science community like MATLAB, R or SPSS all implement interfaces to run Python packages within their environments, which made Python the most accessible tool for our pipeline.

Our pipeline is ready for use, however it has some limitations. First, due to limitations in the programming language Python and other programming languages as well, floating point errors might lead to a misclassification of AKI both in the SCr as well as the UO track. These should only affect a very limited amount of edge cases with questionable clinical relevance. Another limitation is that according to KDIGO criteria, in patients under 18 years of age, a reduction in GFR below 35 ml/min per 1.73 $m^2$ is considered an AKI stage 3. Since we currently did not have openly accessible and publishable real world data from individuals younger than 18 years, we did not validate the pipeline on this criterion and exclusively validated it on patients over the age of 18. It has to be noted that using the Cockcroft-Gault equation to calculate GFR can only be an approximation of the real GFR of the patients. At the same time, it is highly dependent on the weight that has been used for input, e.g. adjusted or real bodwy weight. There is currently no standard of sample sizes that should be used for testing in software development. We decided to use 15 patients in total, which accumulated to 2651 hours in the ICU. So in total, both pyAKI and the physicians labeling the data saw 2651 data points.

Additionally, we implemented extensive testing of our tools on synthetic data. These tests are included in our official release of the toolbox. However, this lack of standardization in software testing in this area can be seen as a limiation to our approach. Finally, heterogeneity in electronic health record databases may require modifications of the data structure before pyAKI can be implemented in some cases. These modifications might introduce bias to the data analysis that are beyond the capabilities of pyAKI. As in most software toolboxes, the quality of the output is still dependent on the quality of the input. In conclusion, pyAKI provides a standardized, validated and openly accessible tool for classifying time series data withAKI labels as defined by KDIGO. These labels achieve and even outperform human level accuracy and can provide a new standard of the definition for AKI in large scale data sets.

## Code availability

The entire source code of pyAKI, as well as the user documentation, along with a full commit history is released to the public under https://www.github.com/aidh-ms/pyAKI.

## Supporting information

**S1 File. Initiation checklist.** The checklist that was used for the initiation of the participating physicians.
(PDF)

## Author Contributions

**Conceptualization:** Christian Porschen, Jan Ernsting, Thilo von Groote.

**Data curation:** Christian Porschen, Jan Ernsting, Till Würdemann, Hendrik Booke, Wida Amini, Ludwig Maidowski, Thilo von Groote.

**Formal analysis:** Christian Porschen, Jan Ernsting, Paul Brauckmann, Raphael Weiss, Till Würdemann.

**Investigation:** Christian Porschen, Jan Ernsting, Tim Hahn.

**Methodology:** Christian Porschen, Jan Ernsting, Paul Brauckmann, Raphael Weiss, Till Würdemann, Hendrik Booke, Wida Amini, Ludwig Maidowski, Benjamin Risse, Tim Hahn.

**Project administration:** Christian Porschen, Jan Ernsting.

**Resources:** Paul Brauckmann.

**Software:** Christian Porschen, Jan Ernsting, Paul Brauckmann, Benjamin Risse.

**Supervision:** Jan Ernsting, Raphael Weiss, Till Würdemann, Benjamin Risse, Tim Hahn.

**Validation:** Christian Porschen.

**Visualization:** Christian Porschen.

**Writing – original draft:** Christian Porschen, Jan Ernsting.

**Writing – review & editing:** Raphael Weiss, Till Würdemann, Hendrik Booke, Wida Amini, Ludwig Maidowski, Benjamin Risse, Tim Hahn, Thilo von Groote.

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
