## [Decision Letter · Decision Letter 0]

15 Aug 2024

PONE-D-24-21062pyAKI - An Open Source Solution to Automated Acute Kidney Injury ClassificationPLOS ONE

Dear Dr. Porschen,

Thank you for submitting your manuscript to PLOS ONE. After careful consideration, we feel that it has merit but does not fully meet PLOS ONE’s publication criteria as it currently stands. Therefore, we invite you to submit a revised version of the manuscript that addresses the points raised during the review process.

We look forward to receiving your revised manuscript.

Kind regards,

Mohammad A. Al-Mamun, PhD

Academic Editor

PLOS ONE

Journal Requirements:

3.Thank you for stating the following financial disclosure: "C.P. - is supported by the Deutsche Forschungsgemeinschaft (DFG, German Research

Foundation) – 493624047 (Clinician Scientist CareerS M¨unster).

T.vG. is supported by the Deutsche Forschungsgemeinschaft (DFG, German Research

Foundation) – 493624047."

4. Thank you for stating the following in the Acknowledgments Section of your manuscript: "Christian Porschen is supported by the Deutsche Forschungsgemeinschaft (DFG, German Research

Foundation) – 493624047 (Clinician Scientist CareerS M¨unster).

Thilo von Groote is supported by the Deutsche Forschungsgemeinschaft (DFG, German Research

Foundation) – 493624047."

Please remove any funding-related text from the manuscript and let us know how you would like to update your Funding Statement. Currently, your Funding Statement reads as follows: "C.P. - is supported by the Deutsche Forschungsgemeinschaft (DFG, German Research

Foundation) – 493624047 (Clinician Scientist CareerS M¨unster).

T.vG. is supported by the Deutsche Forschungsgemeinschaft (DFG, German Research

Foundation) – 493624047."

5. Please note that your Data Availability Statement is currently missing the repository name. If your manuscript is accepted for publication, you will be asked to provide these details on a very short timeline. We therefore suggest that you provide this information now, though we will not hold up the peer review process if you are unable.

6. Please note that in order to use the direct billing option the corresponding author must be affiliated with the chosen institute. Please either amend your manuscript to change the affiliation or corresponding author, or email us at plosone@plos.org with a request to remove this option.

Reviewers' comments:

Reviewer's Responses to Questions

**Comments to the Author**

1. Is the manuscript technically sound, and do the data support the conclusions?

Reviewer #1: Partly

Reviewer #2: Yes

2. Has the statistical analysis been performed appropriately and rigorously? 

Reviewer #1: No

Reviewer #2: Yes

3. Have the authors made all data underlying the findings in their manuscript fully available?

Reviewer #1: Yes

Reviewer #2: Yes

4. Is the manuscript presented in an intelligible fashion and written in standard English?

Reviewer #1: Yes

Reviewer #2: Yes

5. Review Comments to the Author

Reviewer #1: The authors introduced an open-source solution using data analysis in order to improve identification and classification of AKI.

1. The abstract has to better explain the methods employed, and the results of the study for the reader to better understand what are the key aspects of this study.

2. The introduction is too extensive; please reduce highlighting the growing role of large databases in determining the incidence and prognosis of AKI and evaluating initiatives to improve the quality of care in AKI. Using examples, illustrate this use of routinely collected health data and discuss the strengths, limitations, and implications for researchers and clinicians.

3. The approach for estimate baseline is incorrect, measures to estimate baseline SCr can either underestimate or overestimate AKI incidence, which affect outcomes associated with presumed AKI. In this setting, the Kidney Disease Improving Global Outcomes (KDIGO) guidelines suggest using a SCr computed from the Modification of Diet in Renal Disease (MDRD) formula, assuming an estimated glomerular filtration rate (eGFR) of 75 ml/min/1.73 m2, another approach that could be used is the European Renal Best Practice guideline that recommend the use of first SCr at admission. The former assumes that there is a relatively low rate of CKD, while the latter assumes that AKI does not occur before hospitalization.

4. I recommend that an analysis should be run comparing the diagnostic performance between data analysis model and human, using AUC-ROC comparing the different diagnostic criteria used.

Reviewer #2: Reviewer Comments:

Thank you for the opportunity to review this important and practical work. I commend the authors in aiming to address an overdue need to standardize AKI diagnosis across all the healthcare system.

Abstract:

(1) Authors state: “a negative impact on workload and study quality.” Workload for whom (clinicians, medical coders, patients)? Study quality-past and future universal AKI diagnostic criteria to be applied more accurately? Please edit for clarification

(2) Material and Methods, author’s state, “we defined a standardized data model.” Use of defined is misleading- I believe the intent is “we constructed” consider revision for clarity.

(3) Materials and Methods, authors state pyAKI’s performance was performed and later state internal validation as well but do not state here “how” validation was methodologically performance.

(4) Results, Accuracy of 1.0 was stated, what value or metric was used to state the performance result(s).

(5) Discussion, authors state, “it provides a standardized data model and a comprehensive solution for consistent diagnosis,” to be utilized where (ex: during hospitalization, outpatient and by whom (bedside clinicians in real-time, medical billing coders, researchers, etc.? Please add/edit for context/impact to field of science.

Introduction:

(1) “Acute kidney injury (AKI) is a common organ dysfunction in critically ill patients, affecting up to 50% of all intensive care unit (ICU) patients”1,2. “Definitions for AKI changed over the years”3,4. Incomplete sentence- please revise.

(2) “Recently, the definition of AKI has been made easier, by implementing the KDIGO-definition for AKI.” Please add context of what was made “easier” and continued challenges despite this to strengthen the ‘need’ for your proposed tool.

(3) “a pipeline implementation for cross-project application defining a common standard for AKI is lacking.” Please elaborate on the ‘negative research effects’ (i.e., consensus, accuracy, validity) this causes and “what” impact this has on data science and future tool development.

(4) “The entire software we developed is open source and available for download 1. The dataset generated for validation and the software implementation are licensed under an open-source license and publicly available 2.” References provided in manuscript do not correlate with statements proposed.

Materials and Methods:

(1) “Dialysis was also considered for classifying AKI. Any use of dialysis was classified as KDIGO stage 3. Definition of anuria varies in the literature.” Fragmented sentences, consider revising for clarity.

(2) “Users can decide for themselves whether an ideal body weight or adjusted body weight should be used.” Limitations of the Cockcroft-Gault equation and “which weight” should be used remains an area of controversy. The modified formula using adjusted body weight if a height is provided in the proposed tool is certainly beneficial to improve eGFR accuracy. However, please further describe how the eGFR accuracy may be influenced by this somewhere within the manuscript.

(3) Section 2.2, “As the input, the output also is formatted as a time series.” Please revise for clarification. Consider relocating to the first sentence of section 2.2 to state time series formatting for the input and output data values.

(4) “AKI stages are evaluated at each point in time, in the case of our validation experiment, meaning at each hour of the ICU stay.” Awkwardly stated, consider ‘In our validation experiment, AKI stages are assessed at each point in time, specifically at every hour during the ICU stay.’

2.3 Software Development:

(1) I cannot comment on this section as I am not qualified to evaluate.

2.4 Validation Data Set:

(1) “14 patients that had an AKI and one patient not fulfilling KDIGO criteria to serve as control.” The extremely small sample size is a major limitation to serve as a control in the validation process. Understandably, this is due to the limitations of the MIMIC-IV dataset. However, I encourage the authors to consider additional mitigation strategies to improve the sample size to better support the validation accuracy. (Ex: synthetic knowledge synthesis strategies)

3.2 Accuracy:

(1) Please state what an ‘acceptable accuracy standard’ threshold is within the manuscript.

(2) “after comparing to the labels generated by the senior physicians is depicted in table 3.3”. I do not see a table labeled 3.3 in the manuscript (table 2?)

3.5 MIMIC Comparison:

(1) Table 3, for clarification, please depict “what the numeric values in the MIMIC and pyAKI columns are referring to.” (quantity of labels?)

Discussion:

(1) “PyAKI outperformed the accuracy of a group of physicians on the intern level.” Intern level has not been described; I’m assuming you are referring to junior physicians?

(2) “On one occasion, due to a transmission error, a patient was classified to not having AKI at all by the physician’s classification, when in fact, pyAKI and the experts classification agreed, that the patient had AKI. Such minor mistakes, especially due to wrong transmission of single-category AKI determination to the overall AKI stage are typical for human level performance on complex time series data and should be avoidable by using a software pipeline.” Is this necessary to state? Consider relocating to a ‘limitations’ section of the manuscript.

6. PLOS authors have the option to publish the peer review history of their article (what does this mean?). If published, this will include your full peer review and any attached files.

Reviewer #1: No

Reviewer #2: No

---

## [Decision Letter · Decision Letter 1]

25 Nov 2024

pyAKI - An Open Source Solution to Automated Acute Kidney Injury Classification

PONE-D-24-21062R1

Dear Dr. Porschen,

We’re pleased to inform you that your manuscript has been judged scientifically suitable for publication and will be formally accepted for publication once it meets all outstanding technical requirements.

Kind regards,

Mogamat-Yazied Chothia, MBChB, FCP(SA), MMed, PhD

Academic Editor

PLOS ONE

Additional Editor Comments (optional):

Reviewers' comments:

Reviewer's Responses to Questions

**Comments to the Author**

1. If the authors have adequately addressed your comments raised in a previous round of review and you feel that this manuscript is now acceptable for publication, you may indicate that here to bypass the “Comments to the Author” section, enter your conflict of interest statement in the “Confidential to Editor” section, and submit your "Accept" recommendation.

Reviewer #2: All comments have been addressed

Reviewer #3: All comments have been addressed

2. Is the manuscript technically sound, and do the data support the conclusions?

Reviewer #2: Yes

Reviewer #3: Yes

3. Has the statistical analysis been performed appropriately and rigorously? 

Reviewer #2: Yes

Reviewer #3: Yes

4. Have the authors made all data underlying the findings in their manuscript fully available?

Reviewer #2: Yes

Reviewer #3: Yes

5. Is the manuscript presented in an intelligible fashion and written in standard English?

Reviewer #2: Yes

Reviewer #3: Yes

6. Review Comments to the Author

Reviewer #2: Thank you for the opportunity to review this important work. I have no further comments to improve the clarity or context to the topic.

Reviewer #3: concern re Cockcroft-gault formula use

- not that reliable with rapidly changing renal function like AKI

- fairs less well in terms of accuracy of eGFR compared to other tools eg MDRD, CKD-EPI

- weight may difficult to ascertain in critically ill ICU patients without guessing

there are typographical errors which have been attached as comments on the manuscript

7. PLOS authors have the option to publish the peer review history of their article (what does this mean?). If published, this will include your full peer review and any attached files.

Reviewer #2: **Yes: **Todd Brothers, Pharm D, BCCCP, BCPS

Reviewer #3: **Yes: **Zaheera Cassimjee

---

## [Editor Report · Acceptance letter]

13 Dec 2024

PONE-D-24-21062R1 

PLOS ONE

Dear Dr. Porschen, 

I'm pleased to inform you that your manuscript has been deemed suitable for publication in PLOS ONE. Congratulations! Your manuscript is now being handed over to our production team.

Kind regards, 

on behalf of

Prof. Mogamat-Yazied Chothia 

Academic Editor

PLOS ONE